# Evaluation of fecal occult blood testing for rapid diagnosis of invasive diarrhea in young children

**David A. Kwasi**[1,2]**, Pelumi D. Adewole**[1]**, Olabisi C. Akinlabi**[1]**, Stella E. Ekpo**[1]**, Iruka N. Okeke**[1] *

**1** Department of Pharmaceutical Microbiology, Faculty of Pharmacy, University of Ibadan, Ibadan, Oyo State, Nigeria, **2** Department of Pharmaceutical Chemistry, Faculty of Pharmacy, University of Ibadan, Ibadan, Oyo State, Nigeria

* iruka.n.okeke@gmail.com

**Data Availability Statement:** Sequence data were submitted to ENA and are available from ENA https://www.ebi.ac.uk/ena/browser/home and Genbank https://www.ncbi.nlm.nih.gov/genbank/

## Abstract

Antimicrobials are only indicated in acute childhood diarrhea if it is invasive or persistent. Rapid screening for invasive diarrhea can therefore inform treatment decisions but pathogen identification by culture is slow, expensive and cumbersome. This study aimed to assess the diagnostic utility of stool microscopy and immunochromatographic fecal occult blood test (FOBT) kits for identifying invasive or potentially invasive diarrhea in Ibadan, Nigeria. Fecal specimens from 46 children under 5 years old with diarrhea, collected as part of ongoing case-control studies, were subjected to stool microscopy for erythrocytes and leucocytes, and FOBT using the innovator's product and four locally procurable generic immunochromatographic kits, each according to manufacturers' instructions. Stool specimens were cultured for enteric bacterial pathogens using standard procedures. Presumptive pathogen isolates were identified biochemically and by PCR, and then confirmed by whole genome sequencing. *Shigella*, enteroinvasive *Escherichia coli* and *Yersinia*, pathogens that invariably cause invasive diarrhea, were detected in five of 46 specimens. Occult blood detection by microscopy was 55.6% sensitive and 78.4% specific, while the innovator's FOBT product was respectively 62.5% and 81.6% sensitive and specific compared to strict invasive pathogen recovery. Microscopy and FOBT testing were less sensitive in identifying specimens that contained pathogens that do not always elicit invasive diarrhea. Generic FOBT tests compared well with the innovator's product. Microscopy and FOBT testing have some value for delineating likely invasive diarrheas. They could inform treatment and serve as early warning indicators for dysentery outbreaks in resource limited settings. Inexpensive, generic FOBT kits that are locally procurable in Nigeria performed as well as the innovator's product.

## Introduction

Infectious acute diarrhea can be life-threatening in children [1, 2] and hence requires prompt management. Most diarrhea infections are self-resolving with rehydration being the principal

as Bioproject PRJEB8667. All other data are contained in the manuscript.

**Funding:** This work was supported by African Research Leader's Award MR/L00464X/1 to INO. The award is funded by the UK Medical Research Council (MRC) and the UK Department for International Development (DFID) under the MRC/DFID Concordat agreement and is also part of the EDCTP2 programme supported by the European Union. INO is a Calestous Juma Science Leadership Fellow (Award # INV-036234) supported by the Bill and Melinda Gates Foundation. The funders had no role in study design, data collection and analysis, decision to publish, or preparation of the manuscript.

**Competing interests:** The authors have declared that no competing interests exist.

intervention required [3]. Antimicrobials are indicated in invasive and persistent disease and their use is best informed by culture and susceptibility tests. These tests require trained laboratory personnel and take a few days to complete [4]. Inevitably, initial therapeutic choices are not always the most appropriate, which can cause delays in instituting the right treatment or promote antimicrobial use when not indicated. A rapid and cost-effective screening process to identify likely invasive infections at the point-of-care is thus expedient. This is particularly true in Nigeria where diarrhea is an important cause of childhood illness and death and where antimicrobial overuse places the entire population at high risk of the consequences of antimicrobial resistance [5–7]. A range of conditions, including gastrointestinal cancers, malabsorption, abdominal pain, constipation and iron deficiency anaemia, can result in gross or occult blood in stool but the list of conditions includes shigellosis and other enteric invasive infections [8–11].

Unlike culture for invasive pathogens, the presence of blood in stool, a common feature of invasive diarrheas, can be determined within minutes of submission of a specimen and in time to inform initial patient care. Blood in stool is ideally detected by microscopy for erythrocytes and leucocytes but tests designed to detect hemoglobin are commonly used as a diagnostic aid for carcinomas and invasive diarrhea [12]. A meta-analysis by Gill *et al* [13] found that stool microscopy had little utility for identifying invasive diarrheas in resource limited settings, however very few data from outside of high income countries were available for review, likely because microscopy is only accessible in facilities with laboratories and microscopy is also needed to diagnose other common conditions like malaria, parasitic diarrheas, blood and urine infections. Spot tests for hemoglobin do not require microscopes and trained microscopists and are easy to perform. More recently, Bardhan *et. al.* [14] investigated the role of clinical features, stool microscopy, and fecal occult blood testing (FOBT) in distinguishing invasive diarrheas from non-invasive ones in Dhaka, Bangladesh. In that setting, the presence of visible blood in faeces was a reliable indicator of invasive diarrhea. When gross blood or blood cells could not be seen, occult blood was equivalently reliable, unless a test kit with poor sensitivity was employed. Bardhan *et al.* [14] thus found that FOBT was a valuable test in delineating non-bloody diarrhea in Dhaka, and this was comparable to fecal microscopy outcomes. We observed that several brands of kits for these tests are available on the Nigerian market but not routinely used as diagnostic aids for invasive diarrhea. The aim of this study was to compare the diagnostic efficacy of microscopy and locally procurable rapid fecal occult blood test (FOBT) kits at identifying invasive infantile diarrhea in northern Ibadan, Nigeria.

## Materials and methods

### Ethical considerations

Ethical approval for this work was obtained from the University of Ibadan / University College Hospital (UI/UCH) ethics committee (approval number UI/EC/15/093). Study participants' parents or guardians gave consent for their participation in the study.

### Fecal occult blood test

A total of four FOBT kits were evaluated in this study alongside the innovator's product (Cromatest, Spain) (Table 1). All five kits were lateral flow immunochromatographic kits. Forty-six (46) Fecal specimens from children below 5 years old with diarrhea, being collected as part of a case-control study in our laboratory in Nigeria, were tested according to manufacturers' instructions for each kit. A sample was either considered negative for FOBT if a single band is spotted on the test strip or positive if two bands (one being the control line) was found on the test strip at the end of the testing procedure, in accordance with manufacturer's instructions. Diagnostic test efficacy was computed as described earlier [15].

**Table 1. Preliminary information on the five immunochromatographic FOBT kits investigated.**

| Brand of kit | Country of manufacture | Kit components (Extraction buffer, test strip, reaction cups) | Non-kit components required | Price per unit in 2017 (N) | Storage recommend-ation |
|---|---|---|---|---|---|
| Cromatest* | Spain | 2 | 2 | 600 | 2–30˚C |
| Abon | China | 3 | 2 | 360 | 2–30˚C |
| Diaspot | Indonesia | 3 | 2 | 360 | 2–30˚C |
| LabACON | China and Canada | 3 | 2 | 280 | 2–30˚C |
| MICROPOINT | USA | 2 | 3 | 280 | 2–30˚C |

*Innovator's product

### Stool microscopy testing and culture

Fecal microscopy for leukocytes and erythrocytes was done using wet mount method [16]. Bacterial culture of stool specimens were performed as described elsewhere [17]. Briefly, stool specimens were plated on MacConkey, Eosin Methylene blue (EMB) and Xylose Lysine Deoxycholate (XLD) agar and incubated at 37 ˚C. Specimens were also enriched for Salmonella, using Selenite broth, followed by sub-culture onto XLD. Up to ten distinct colonies of lactose and non-lactose fermenters were picked from MacConkey and EMB plates while black centered colonies with slightly red edges were picked from XLD. Biochemical identification of isolates was carried out using Microbat 12B, 12E and 24E kits. Molecular identification of pathogen subtypes were performed by polymerase chain reaction (PCR), as described previously [18, 19]. Briefly, isolate DNA was extracted aseptically using the Wizard Genomic Extraction kit (Promega). PCR was performed for enteropathogenic, enterotoxigenic, enteroinvasive, enteroaggregative and Shiga-toxin-producing *E. coli*, and for *Salmonella enterica* using the methods described earlier [20, 21]. Identified pathotypes were confirmed by whole genome sequencing using Illumina platform. Raw reads quality check, assembly, assembly quality check and speciation was done according to Akinlabi *et al.*, 2023. Sequence data were submitted to ENA and are available from ENA https://www.ebi.ac.uk/ena/browser/home and Genbank https://www.ncbi.nlm.nih.gov/genbank/ as Bioproject PRJEB8667.

### Data analysis

The sensitivity, specificity, positive and negative predictive values of microscopy for leukocytes, for erythrocytes and FOBT using the innovator's immunochromatographic kit for identifying pathogens that are invariable invasive (*Shigella*, enteroinvasive *E. coli* and *Yersinia*) and sometimes invasive (*Salmonella* and enteroaggregative *E. coli*) were computed as listed below. We additionally compared these outcomes of the generic FOBT tests on the Nigerian market to the innovator's product. Statistical testing was performed by the Fisher's Exact Test in EpiInfo Software.

Test specificity, sensitivity, positive predictive values, and negative predictive values were computed using the following formulae [15]:

$$Specificity = \frac{True\ negatives}{True\ negative + False\ positive} \times 100$$

$$Sensitivity = \frac{True\ positive}{True\ positive + False\ negative} \times 100$$

$$\text{Positive predictive value} = \frac{\text{True positive}}{\text{True positive} + \text{False positive}} \times 100$$

$$\text{Negative predictive value} = \frac{\text{True negative}}{\text{True negative} + \text{False negative}} \times 100$$

## Results

### Utility of stool microscopy and FOBT immunochromatographic testing for identifying likely invasive diarrheas

The results of microscopy, FOBT testing with all kits, as well as pathogen culture and identification are contained in Table 2. While our stool pathogen screening was not exhaustive, we aimed to identify invasive bacterial pathogens for which antibacterial therapy would be appropriate. A very broad range of agents were identified, including *Shigella* and *Yersinia*, which invariably result in invasive infections as well as *Salmonella*, enteroaggregative *E. coli* (EAEC) and cell-detaching *E. coli* (CDEC) which may or may not, as a result of pathogen as well as host factors [21–24]. Enterohemorrhagic and enteropathogenic *E. coli* were sought but not detected in the study specimens. One specimen contained enterotoxigenic *E. coli* (ETEC), a well-characterized non-invasive pathogen.

Of the bacterial pathogens sought, *Shigella* and enteroinvasive *E. coli* (EIEC) and *Yersinia*, spp. invariably produce dysenteric or invasive infections, *Salmonella* often does, while enterotoxigenic (ETEC) and enteropathogenic *E. coli* (EPEC) typically do not. The EAEC pathotype is highly heterogenous and believed to comprise invasive and non-invasive strains however most isolates are believed to be non-invasive. It is not known whether or CDEC are invasive but they express alpha-haemolysin and cytolethal distending factor toxins [21, 23, 24]. In this study, 21 of the 46 specimens yielded at least one bacterial pathogen, with five of the children from which these specimens were derived suffering mixed infections. Potentially invasive diarrheal pathogens were detected in 21 of the 46 samples tested and five of these contained a pathogen that invariably produces invasive diarrhea, that is *Shigella*, EIEC or *Yersinia*. Detectable blood in stool was associated with these strict invasive pathogens. Occult blood detected by microscopy, as defined by $> 2$ erythrocytes and $\geq 1$ leucocytes per field, was seen in four of the five samples from which an invariably invasive bacterial pathogen was cultured and seven of the specimens where no invasive bacterium was detected ($p = 0.009$). The sensitivity and specificity of erythrocytes, leucocytes and occult blood by microscopy for detecting these pathogens were 55.6% and 78.4% (Table 3). When we assessed the performance of the innovator FOBT test kits, we found that detection of hemoglobin using the Cromatest kit was similarly associated with recovery of a strict invasive pathogen ($p = 0.007$) and had a sensitivity and specificity of 62.5% and 81.6% compared to strict invasive pathogen recovery (Table 3). Neither microscopy for occult blood nor haemoglobin detection by FOBT was associated with recovery of the full list of potentially invasive pathogens (*Shigella*, EIEC, *Yersinia*, *Salmonella*, EAEC or CDEC). As shown in Table 3, while specificities for collectively predicting the presence of these pathogens were also above 70%, the sensitivities were under 35%.

### Diagnostic efficacy of FOBT kits evaluated

The innovator product and four other FOBT kits tested used an easy-to-follow protocol executable in 4–7 minutes using 2–3 kit and non-kit components, alongside a user-supplied sterile

**Table 2. Microscopy, FOBT outcomes and aetiologic agents.**

| Sample code | Age (Months) | Sex | MICROSCOPY (number of cells per field) | | Occult blood | CROMATEST | Abon | Diaspot | LabACON | MICROPOINT | Pathogens identified |
|---|---|---|---|---|---|---|---|---|---|---|---|
| | | | RBC | WBC | | | | | | | |
| LWD016 | 7 | M | > 20 | > 20 | + | + | + | + | + | + | EIEC |
| CHD048 | 7 | M | > 20 | 8–10 | + | + | + | + | + | + | *Salmonella*. Durham |
| CHD079 | 10 | M | 10–15 | > 20 | + | + | + | + | + | + | Nil |
| CHD043 | 8 | M | 10–12 | 5–6 | + | + | + | + | + | + | *Shigella* serogroup AB, CDEC |
| CHD056 | 21 | F | 6–8 | > 20 | + | + | + | + | + | + | *Yersinia enterocolitica* |
| CHD054 | 19 | M | 2–3 | 1–2 | + | + | + | + | + | + | *Yersinia ruckeri* |
| CHD052 | 20 | M | 1–2 | 4–6 | - | + | + | + | + | + | Nil |
| CHD051 | 20 | M | 2–3 | 2–3 | + | + | - | + | - | + | Nil |
| LLD035 | 6 | M | 1–2 | 2–3 | - | + | + | - | + | + | *Salmonella* Elizabethville |
| CHD067 | 24 | M | 4–5 | 12–15 | + | - | - | - | - | - | EIEC |
| CHD086 | | | 1–2 | >20 | - | - | - | - | - | - | EAEC |
| CHD087 | 19 | M | 2–3 | >20 | + | - | - | - | - | - | Nil |
| LLD039 | | | 1–2 | 10–20 | - | - | - | - | - | - | Nil |
| CHD101 | 24 | M | 2–3 | >20 | + | - | - | - | - | - | Nil |
| LWD042 | | | 0–2 | 15–20 | - | - | - | - | - | - | *Salmonella* Riverside |
| CHD102 | 3.5 | M | 0–2 | >20 | - | + | + | + | + | + | EAEC |
| CHD103 | 21 | M | 2–3 | >20 | + | - | - | - | - | - | Nil |
| CHD104 | 28 | M | 1–2 | 8–10 | - | - | - | - | - | - | Nil |
| CHD105 | 12 | M | 3–4 | >20 | + | + | + | + | + | + | Nil |
| CHD106 | 20 | F | 0–2 | >20 | - | + | + | + | + | + | Nil |
| CHD107 | 20 | F | 2–3 | >20 | + | - | - | - | - | - | Nil |
| CHD108 | 24 | M | 1–2 | 15–20 | - | - | - | - | - | - | EAEC |
| CHD109 | 7 | M | 1–2 | 15–20 | - | - | - | - | - | - | Nil |
| MND006 | | | 0–2 | 8->20 | - | - | - | - | - | - | CDEC (1), ETEC-ST (1) |
| CHD010 | 19 | M | | | | | | | | | |
| CHD045 | 4 | M | | | | | | | | | |
| CHD061 | 9 | F | | | | | | | | | |
| CHD008 | 7 | F | 0–2 | 0–8 | - | - | - | - | - | - | *Y. ruckeri* (1) |
| CHD011 | 3.5 | F | | | | | | | | | Nil |
| CHD013 | 15 | | | | | | | | | | Nil |
| CHD014 | 11 | F | | | | | | | | | EAEC |
| CHD015 | 9 | | | | | | | | | | Nil |
| CHD049 | 36 | M | | | | | | | | | Nil |
| CHD050 | 0.5 | M | | | | | | | | | Nil |
| CHD057 | 8 | M | | | | | | | | | Nil |
| CHD060 | 4 | F | | | | | | | | | Nil |
| CHD062 | 19 | M | | | | | | | | | Nil |
| CHD075 | 14 | M | | | | | | | | | Nil |
| CHD085 | 7 | M | | | | | | | | | Nil |
| LKD008 | 12 | F | | | | | | | | | EAEC |
| LLH031 | 3 | F | | | | | | | | | EAEC |
| LWD010 | 19 | M | | | | | | | | | EAEC |
| LWD015 | 14 | F | | | | | | | | | EAEC |
| LWD029 | 10 | F | | | | | | | | | EAEC |
| LWD030 | 12 | F | | | | | | | | | EAEC |
| LWD11 | 12 | M | | | | | | | | | EAEC |

EIEC = enteroinvasive *E. coli*, EAEC = enteroaggregative *E. coli*, ETEC = enterotoxigenic *E. coli*, CDEC = cell-detatching *E. coli*.

**Table 3. Diagnostic efficacy of microscopy and immunochromatographic testing with the innovator kit compared to pathogen recovery.**

| Compared to recovery of | Test | Positives | Negatives | Sensitivity (%) | Specificity (%) | Positive predictive value (%) | Negative predictive value (%) |
|---|---|---|---|---|---|---|---|
| Strictly invasive pathogens: *Shigella*, EIEC, *Yersinia* | Microscopy | 13 | 33 | 55.56 | 78.38 | 38.46 | 87.88 |
| | Cromatest | 12 | 34 | 62.50 | 81.58 | 41.67 | 96.88 |
| Potentially invasive pathogens: *Shigella*, EIEC, *Yersinia*, *Salmonella*, EAEC, CDEC | Microscopy | 13 | 33 | 28.57 | 72.00 | 46.15 | 54.55 |
| | Cromatest | 12 | 34 | 33.33 | 84.00 | 63.64 | 95.45 |

swab sticks and stool collection containers (Table 1). The innovator's product had slightly fewer components and therefore could be considered less complex than the other tests but was priced at almost twice the average cost of the four locally procurable test kits (Table 1). All tests gave control bands for all tests, so that none of the test strips used had to be invalided. As shown in Fig 1, a positive result was easy to call for each kit. Twelve (12) of 46 (26.1%) specimens examined gave positive FOBT outcomes with the innovator's kit (Cromatest), eight of which were also positive for stool erythrocytes (Table 2). All kits evaluated had comparable specificity with the innovator product (Cromatest) (Table 4), with generic kits yielding slightly different results in two specimens that had fewer erythrocytes (<3) per field as shown in Table 2. As shown in Table 4, two of the cheaper products that are readily available in Nigeria (Diaspot and Micropoint) performed as well as the innovator product. All five kits were as specific as the innovator product (Cromatest) with Diaspot and Micropoint exhibiting higher sensitivity (Table 4).

## Discussion

Unless gross blood or mucus are seen in stools, delineating childhood diarrheas as potentially dysenteric gastrointestinal episodes, which require antimicrobials, is difficult on the basis of clinical signs and symptoms alone. A conventional approach to rapidly detect occult

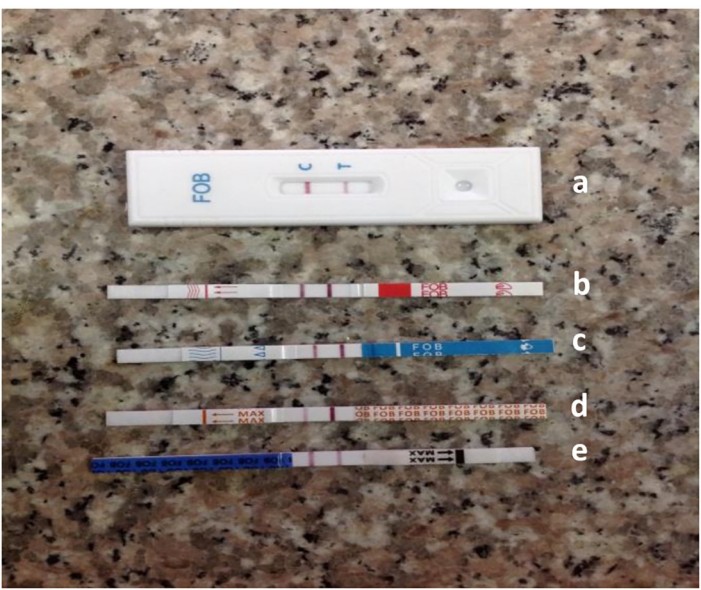

**Fig 1. Positive test results for the five FOBT kits tested.** (a) CROMATEST, (b) Abon, (c) Diaspot, (d) LabACON and (e) Micropoint. On each test strip, the control band is marked 'C' and the test band 'T'.

**Table 4. Comparison of diagnostic potentials of innovator and generic FOBT kits.**

| FOBT kit | Positives | Negatives | Sensitivity (%) | Specificity (%) | Positive predictive value (%) | Negative predictive value (%) |
|---|---|---|---|---|---|---|
| Abon | 11 | 35 | 91.67 | 100.00 | 91.67 | 100.00 |
| Diaspot | 12 | 34 | 100.00 | 100.00 | 100.00 | 100.00 |
| LabACON | 11 | 35 | 91.67 | 100.00 | 100.00 | 100.00 |
| Micropoint | 12 | 34 | 97.14 | 100.00 | 97.14 | 100.00 |

blood in stool by microscopy, which can be performed in under an hour even though it is tedious, requires trained personnel and uses resources also used for malaria testing in our setting. Culture can identify invasive bacterial pathogens at slightly higher throughput but takes 2–3 days and additionally also requires skilled-labor. Fecal occult blood testing (FOBT) by immunochromatographic methods can be performed rapidly at the point of care but has not heretofore been evaluated for utility in delineating invasive childhood diarrheas in our setting.

In this study, as have others, we found that microscopy for occult blood has good but less-than-perfect sensitivity and high specificity for identifying diarrheas associated with strict invasive pathogens, and performed similarly to the innovator's immunochromatographic FOBT test, which can be used at the point of care. In the case of *Salmonella*, EAEC and CDEC, which can cause invasive or non-invasive diarrheas, both blood-in-stool tests were inadequately sensitive for predicting the presence of a potentially invasive pathogen but this is likely because these pathogens were only producing invasive disease some of the time.

The innovator's FOBT immunochromatographic kit is intermittently available in Nigeria and costlier than competitor products. We identified four generic FOBT test kits, which could be procured within Ibadan, Nigeria, a non-coastal city, without an international airport. In our evaluation, all five kits gave similar results and compared reasonably well with expert microscopy.

A large number of potentially invasive bacterial pathogens was detected by culture in the specimens and in most cases, a positive FOBT result was obtained with any kit. Even though we did not seek *Campylobacter* spp. or protozoal parasites that can elicit blood in stool, significantly higher number of specimens with positive FOBT outcomes had strict invasive pathogens. Based on our findings, occult blood tests can play a role in identifying diarrheal cases requiring antibacterial therapy when more in-depth lab testing is unavailable. On the average, it takes 4–7 minutes to conduct test per specimen using any of the 5 FOBT kits. In 2017–18, each of these kits cost average of 30 ($ 0.065) at existing exchange rates at the time of kit procurement ($1 at 460.28). The cost and total time taken from sample processing to result demonstrate that FOBT test are fast, cheap, and easy to run.

Our evaluation has some limitations. Because we wished to comparatively evaluate several kits, only a small number of specimens could be screened. The number of specimens that could be evaluated was additionally constrained by Cromatest (the innovator's product) availability and stock outs. Additionally, we did not seek viral or protozoal pathogens, or *Campylobacter* species and many of these can produce invasive disease. However, even with these limitations, the data appear to suggest that fecal occult blood tests have significant value in delineating children with invasive diarrhea, who should receive antibacterial therapy, and that generic products are functionally equivalent for this purpose to the innovator's product. Using these kits at the point-of-care in institutions where laboratory testing would normally be unavailable could help to improve patient care and contain antimicrobial resistance.

## Conclusion

Microscopy and FOBT kits are rapid, cost effective and valuable screening processes for quick diagnosis of diarrhea. Stool microscopy can be performed in facilities that have this resource and where it is not available, pediatric specimens can be subjected to FOBT hemoglobin testing at the point-of-care. In this regard, inexpensive, locally available kits perform similarly to the more difficult-to-procure innovator's product. If used routinely, blood-in-stool evaluations could avoid unnecessary empiric antimicrobial prescription and could also be an early warning indicator for outbreaks due to invasive pathogens.

## Acknowledgments

We thank UNITECH laboratory Services for their assistance in market surveys for FOBT products in Nigeria, Abiodun Oyerinde and Emmanuel Bamidele for technical assistance and Aaron Oladipo Aboderin for helpful discussions.

## Author Contributions

**Conceptualization:** Iruka N. Okeke.

**Data curation:** David A. Kwasi, Iruka N. Okeke.

**Formal analysis:** David A. Kwasi.

**Funding acquisition:** Iruka N. Okeke.

**Investigation:** David A. Kwasi, Pelumi D. Adewole, Olabisi C. Akinlabi, Stella E. Ekpo.

**Methodology:** Pelumi D. Adewole, Stella E. Ekpo, Iruka N. Okeke.

**Project administration:** David A. Kwasi, Iruka N. Okeke.

**Supervision:** Iruka N. Okeke.

**Validation:** Olabisi C. Akinlabi.

**Writing – original draft:** David A. Kwasi, Iruka N. Okeke.

**Writing – review & editing:** Pelumi D. Adewole, Olabisi C. Akinlabi, Stella E. Ekpo, Iruka N. Okeke.

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
