## [Decision Letter · Decision Letter 0]

25 Apr 2023

PGPH-D-23-00166

Evaluation of Fecal Occult Blood Testing Kits for Rapid Point-of-Care Diagnosis of Invasive Diarrhea in Young Children

Dear Dr. Okeke,

Thank you for submitting your manuscript to PLOS Global Public Health. After careful consideration, we feel that it has merit but does not fully meet PLOS Global Public Health’s publication criteria as it currently stands. Therefore, we invite you to submit a revised version of the manuscript that addresses the points raised during the review process.

We look forward to receiving your revised manuscript.

Kind regards,

Sara Suliman

Academic Editor

Journal Requirements:

1. Please provide separate figure files in .tif or .eps format only and remove any figures embedded in your manuscript file. Please also ensure that all files are under our size limit of 10MB.

2. Figure 1: Please confirm (a) that you are the photographer; or (b) provide written permission from the photographer to publish the photo(s) under our CC-BY 4.0 license.

Additional Editor Comments (if provided):

Reviewers' comments:

Reviewer's Responses to Questions

**Comments to the Author**

1. Does this manuscript meet PLOS Global Public Health’s publication criteria? Is the manuscript technically sound, and do the data support the conclusions? The manuscript must describe methodologically and ethically rigorous research with conclusions that are appropriately drawn based on the data presented.

Reviewer #1: No

Reviewer #2: Yes

2. Has the statistical analysis been performed appropriately and rigorously?

Reviewer #1: I don't know

Reviewer #2: Yes

3. Have the authors made all data underlying the findings in their manuscript fully available (please refer to the Data Availability Statement at the start of the manuscript PDF file)?

Reviewer #1: Yes

Reviewer #2: No

4. Is the manuscript presented in an intelligible fashion and written in standard English?

Reviewer #1: Yes

Reviewer #2: Yes

5. Review Comments to the Author

Reviewer #1: GENERAL COMMENT

The authors have made an attempt to undertake a technical evaluation of the point of testing kits employed in the evaluation of faecal occult blood in Nigeria. Although the study is important considering issues of weak market surveillance regarding such products in sub-Saharan Afriac in general, there are methodological issues that affect the results and its subsequent interpretation. I have itemized the major issues with the manuscript in its present format in the relevant sections below.

TITLE:

The title is unduly long “Evaluation of Fecal Occult Blood Testing Kits for Rapid Point-of-Care Diagnosis of Invasive Diarrhea in Young Children” should be revised to read something along the lines” Authors should consider revising the title to remove the POC “Evaluation of Fecal Occult Blood Testing Kits for Diagnosis of Invasive Diarrhea in Young Children”.

Abstract

Background: The phrase using faecal microscopy as the gold standard” needs reconsideration. Occult blood test kits are able to detect microscopic levels of free haemoglobin which cannot be detected by light microscopy. Light microscopy is only able to detect intact red cells and in possibly red cell casts. Therefore, the choice of light microscopy as the gold standard to compare other test platforms (FOBT) which detect haemoglobin that is not within the detection threshold of light microscopy raises methodological concerns. Authors should consult a similar study by Bardhan et al which used Isolation of faecal enteropathogens served as the gold standard, when comparing the performance of microscopy and FOBT. https://doi.org/10.1080/003655200750024533

Introduction

In line 65 – 66, authors acknowledge that the focus of faecal microscopy and FOBT are separate when they wrote “Blood in stool is ideally detected by microscopy but tests designed to detect haemoglobin are commonly used as a diagnostic aid for carcinomas and invasive diarrhea”. Therefore, one expects these methodological differences to inform the methods and standards selected.

MATERIALS AND METHODS

Ethical considerations

Guardians give assent on the behalf of minors, but not consent; please revise.

FAECAL OCCULT BLOOD TEST

There are currently three types of FOBTs based on different measurement methods: chemical tests, immunochromatographic tests, and DNA tests. These tests differ not only in the detection method but also in their susceptibility to cross-reactions and interfering factors. Authors should enclose the testing principles of each of the FOBT kits used in the present study to enable the reader to situate the results in terms of how specific each test result was.

Stool microscopy

In line with the suggestions above regarding the diagnostic potentials of faecal microscopy and FOBT, authors should consider revising the statement “Fecal microscopy for occult blood (the control assay for the FOBT tests)”

Bacterial culture and biochemical testing

Authors should be specific as the exact culture methods/media employed for the stool cultures. The selection of media for culture is not generic as it is influenced by the organism being suspected. Therefore, just broadly saying “Bacterial culture of stool specimens was performed using standard methods as described by Murray et al., 1995” is not sufficient in this case. Growth conditions for salmonella is not the same as vibro cholera etc.

RESULTS & DISCUSSION

In view of the choice of gold standard as raised in the above discussion, I am unable to comment on the PPV, NPV etc. results. Authors should consider revising the results and discussion sections once the above recommendations are affected.

Reviewer #2: Kwasi and colleagues compared the performance of locally available test kits from different manufacturers for the diagnosis of invasive diarrhea in young children

They showed that the performance of the test kits was comparable and could be used in settings where the skills and resources for more advanced laboratory tests are not available.

Overall, it was a relevant topic worth investigation and could lead to reduction in morbidity and mortality from diarrhea in children

Below are some comments for the authors to consider addressing to improve the manuscript

1. Basic demographics of the participants should be included

2. The methods could be expanded for easy understanding. Currently, in most cases, the authors refer to other published manuscripts. It would be beneficial if the authors can at least briefly describe some of the methods e.g…how the diagnostic test efficacy was determined…how the culture for bacterial pathogens was done…description is currently very scanty.

3. In table 1, I was not sure the relevance of presenting Kit components and no kit components results. The relevance and how these could have impacted the results obtained from each kit should be stated.

4. Is there any relationship between the presence of RBC/WBC in stools and detection of invasive/non invasive bacteria?

5. Figure 1 and description of figure legend not very visible/clear to me. The strips should be clearly labelled and positive and control test bands identified for each strip. Also if possible include a negative test (for each kit).

6. UI/UCH should be written in full in line 80

7. Expand columns for kits in table 2 for row 1 to show full text

Typos

1. Line 53: Life-threatening

6. PLOS authors have the option to publish the peer review history of their article (what does this mean?). If published, this will include your full peer review and any attached files.

**Do you want your identity to be public for this peer review?** For information about this choice, including consent withdrawal, please see our Privacy Policy.

Reviewer #1: **Yes: **Patrick Adu (Ph.D)

Reviewer #2: **Yes: **Muki Shey

---

## [Editor Report · Decision Letter 1]

21 Jun 2023

Evaluation of Fecal Occult Blood Testing for Rapid Diagnosis of Invasive Diarrhea in Young Children

PGPH-D-23-00166R1

Dear %TITLE% Okeke,

We are pleased to inform you that your manuscript 'Evaluation of Fecal Occult Blood Testing for Rapid Diagnosis of Invasive Diarrhea in Young Children' has been provisionally accepted for publication in PLOS Global Public Health.

Best regards,

Sara Suliman

Academic Editor